# In Vitro Modeling of Diabetes Impact on Vascular Endothelium: Are Essentials Engaged to Tune Metabolism?

**DOI:** 10.3390/biomedicines10123181

**Published:** 2022-12-08

**Authors:** Alexander V. Vorotnikov, Asker Y. Khapchaev, Alexey V. Nickashin, Vladimir P. Shirinsky

**Affiliations:** National Medical Research Center of Cardiology Named after Academician E.I. Chazov, 121552 Moscow, Russia

**Keywords:** vascular endothelial cells, type 2 diabetes, hyperglycemia, hyperlipidemia, free fatty acids, carnitine, AMP-activated protein kinase, oxidative stress, mitochondria, uncoupling protein 2

## Abstract

Angiopathy is a common complication of diabetes mellitus. Vascular endothelium is among the first targets to experience blood-borne metabolic alterations, such as hyperglycemia and hyperlipidemia, the hallmarks of type 2 diabetes. To explore mechanisms of vascular dysfunction and eventual damage brought by these pathologic conditions and to find ways to protect vasculature in diabetic patients, various research approaches are used including in vitro endothelial cell-based models. We present an analysis of the data available from these models that identifies early endothelial cell apoptosis associated with oxidative stress as the major outcome of mimicking hyperglycemia and hyperlipidemia in vitro. However, the fate of endothelial cells observed in these studies does not closely follow it in vivo where massive endothelial damage occurs mainly in the terminal stages of diabetes and in conjunction with comorbidities. We propose that the discrepancy is likely in missing essentials that should be available to cultured endothelial cells to adjust the metabolic state and withstand the immediate apoptosis. We discuss the role of carnitine, creatine, and AMP-activated protein kinase (AMPK) in suiting the endothelial metabolism for long-term function in diabetic type milieu in vitro. Engagement of these essentials is anticipated to expand diabetes research options when using endothelial cell-based models.

## 1. Introduction

Vascular endothelium is the primary target of hyperlipidemia, hyperglycemia, and insulin in metabolic disorders and diabetes, leading to endothelial dysfunction and cardiovascular complications at various stages of pathological changes [1,2,3]. Being a frequent comorbidity of atherosclerosis, diabetes accelerates its progression by damaging vital biomolecules through oxidation, glycation, and a plethora of uncontrolled downstream reactions, stimulates inflammation and associated vascular injury leading to atherothrombosis, thus contributing to increased cardiovascular mortality worldwide [4,5,6]. Understanding molecular features and mechanisms that underlie altered physiology of endothelial cells caused by high levels of ambient glucose or free (nonesterified) fatty acids (FFA) is imperative to develop preventive approaches and targeted therapy of both the metabolic and cardiovascular threats. However, the great obstacle to directly studying unique responses of vascular endothelium in vivo or in situ is its fragile, wafer-thin, and hard to reach nature. For this reason, these responses are usually assessed indirectly via changes of the vascular wall reactivity to endothelium-derived substances should they change under disease conditions or those modelled [7]. A proven enhancement to in vivo/in situ research tools is in vitro endothelial cell-based models [8]. Endothelial cells are isolated from tissue biopsies, cultured to propagate, and provide means to manipulate gene expression, uniformly apply stimuli, and enable (patho)physiologic and microstructural studies. The cell-based models provide full control to a researcher and allow the mechanistic studies that are hard to perform in the in vivo setting.

The critical issue that arises in such an approach is whether the culture conditions adequately match those in situ [9]. Considerable efforts have been made to develop culture media suitable for propagation of endothelial cells and resulted in the design of an apparent endothelial basal medium (EBM) prototype also known as MCDB-131 [10]. To enable cell growth, EBM is further furnished with a defined set of growth factors, hormones, vitamins, and other additives to obtain the so-called endothelial growth medium (EGM). Still, EGM contains up to 5% fetal serum to deliver yet uncharacterized factors required for endothelial cells. Compared to the original endothelial growth media, such as M199, DMEM, MEM, F12, etc., supplemented with 10–20% fetal serum, EGM has more defined composition. Several media are now available from various vendors that implement the same or similar EBM/EGM strategy [11]. The most popular vendors include Lonza (Switzerland), Cell Applications (USA), PromoCell (Germany), and ScienCell (USA). The extensive use of low-serum EGM by many laboratories across the globe allowed standardizing the endothelial cell culture conditions and ensured a cross-comparison of the results from various fields of endothelial research.

In particular, metabolic aberrations typical for type 2 diabetes (T2D) have been reproduced in endothelial cell culture in multiple studies in order to reveal how these pathologic conditions affect vascular endothelium. They included exposure of endothelial cells to high environmental glucose or lipid levels typical for T2D progression and implicated in endothelial dysfunction. High content of fatty acids with low carbon number and few/no double bonds in plasma triglycerides and phospholipids are associated with increased risk of diabetes [12]. Palmitic acid, an abundant fatty acid of this type, has been widely used to model hyperlipidemia of prediabetic/diabetic obesity.

In vitro studies promptly identified endothelial apoptosis and cell death as the most common and immediate cell response to high glucose (HG) or palmitate (Figure 1). Overall, these findings have led to a general view on apoptosis as an ultimate outcome of diabetic complications with increased oxidative stress and endothelial dysfunction being just an intermediate step [13,14,15,16]. Yet, it should be stressed that no massive apoptosis of vascular endothelial cells occurs under such metabolic conditions in vivo unless accompanied by inflammation [17,18], atherosclerosis [19], or other diseases [18]. This discrepancy likely stems from the suboptimal composition of endothelial growth media that failed to support endothelium in stringent environment such as that developed in blood in the course of T2D progression.

Below, we review evidence that some essential substances might be missing from the endothelial cell growth media, especially those substances, which are normally produced by other tissues, carried by the blood, and uptaken by vascular endothelium to sustain viability. We suggest that endothelial cells possess an intrinsic mechanism that acts in concert with these essential substances to sense and respond to stressful metabolic conditions in order to tune up metabolism and withstand apoptosis. We discuss metabolic aspects and the key molecules, deteriorated activity of which may lead to vascular endothelium dysfunction.

## 2. Hyperglycemia Sets Off Multiple Routes to Endothelial Dysfunction and Apoptosis

Table A1 presents the summary of in vitro studies using cultured endothelial cells treated by HG to mimic hyperglycemia. Collectively, they suggest that hyperglycemia induces cell apoptosis via increased reactive oxygen species (ROS) and endothelial nitric oxide synthase (eNOS) uncoupling, oxidative stress, and activation of autophagy (Figure 1). In addition, peroxynitrite accumulation and nitrosative stress accompany oxidative stress, and this mechanism appears important for endothelium since it involves altered eNOS activity. In endothelial cells, insulin signaling that targets eNOS activity through the regulatory phosphorylation of this enzyme also appears to be impaired in hyperglycemia, although this issue was not thoroughly investigated.

Increased ROS generation by endothelial cells is consistently observed in response to both HG and FFA treatment (see below). ROS act as a common upstream event to initiate an array of responses converging into several pathogenic mechanisms [20]. In response to glucose overload, ROS are produced by mitochondria, NADPH-oxidases (NOX), and uncoupled eNOS to trigger DNA double-strand breaks in the nucleus and activation of poly(ADP-ribose) polymerase 1 (PARP-1). PARP-1 depletes intracellular NAD^+^ and inhibits a key glycolytic enzyme, glyceraldehyde-3-phosphate dehydrogenase, by its polyADP-ribosylation. This causes upstream accumulation of early glycolytic intermediates, which are diverted to activate sorbitol, hexosamine, diacylglycerol/PKC, and advanced glycation end-products (AGE) pathways leading to apoptosis [20]. Depletion of NAD^+^ also results in SIRT1 inhibition and reduction in AMP-activated protein kinase (AMPK) activity, consistent with lower AMPK activity in insulin-resistant and obese subjects [21,22]. Pharmacological activation of AMPK by AICAR or constitutively active AMPK alleviates the HG-induced oxidant stress and endothelial apoptosis [23].

Measurement of apoptosis in endothelial cell studies requires caution while interpreting the results. Apoptosis is usually assessed indirectly using [^3^H]thymidine incorporation, DNA laddering, TUNEL or annexin-V assays, expression of pro-apoptotic proteins, such as Bax, Bcl-2, or Bak, and caspase-3 activation [24,25,26,27]. All these approaches seem to suffer from a common drawback of inability to catch the maximum cell response and calculate the true extent of apoptosis. Due to intrinsic heterogeneity of cells in culture, duration of apoptosis can vary between the cells [27]. Some of them can quickly die, disintegrate, and escape the assay, whereas others can survive and manifest only modest responses. Another challenge is that apoptosis can occur at low frequency and the dying endothelial cells can be replaced by proliferation and spreading of adjacent cells, so that the culture can be maintained as a whole [27]. In the majority of studies listed in Table A1 and Table A2, cell apoptosis was measured relative to the untreated cells but not as a fraction of maximum response in the same culture. This leaves uncertainty regarding the true extent of apoptosis in endothelial cell models reproducing both hyperglycemia and hyperlipidemia. To obtain a more complete picture of endothelial cell damage brought by these conditions, it is advantageous, in the course of treatment, to supplement the apoptosis measurements with direct visualization of cells or fluorescent viability assays using low magnification light microscopy to enable wide-field inspection of the endothelial monolayer [27,28]. Additionally, time-lapse imaging of endothelial culture allows observing the dynamics of cell monolayer reparation after the loss of individual apoptotic cells or of wide-range deteriorating events.

## 3. Hyperlipidemia Threats Endothelium through Increased FFA Levels

Table A2 presents the summary of in vitro studies performed on cultured endothelial cells exposed to FFA (as FFA: albumin complex) with a particular focus on apoptosis. Similar to hyperglycemia studies, these studies collectively suggest that experimental hyperlipidemia induces endothelial cell apoptosis via increased ROS production and oxidative stress, activation of autophagy, and increased inflammation (Figure 1). Furthermore, frequently reported outcomes are the inhibition of insulin signaling and altered eNOS activity. Notably, most of these studies used relatively low FFA concentrations approximately corresponding to those in healthy human subjects (<0.5 mM), perhaps for the reason that higher FFA concentrations acutely triggered endothelial cell death [28]. Based on 50 years of epidemiological studies, plasma FFA levels of 0.7 mM and higher are considered typical for obesity/disease states whereas ~0.6 mM FFA is approximated as a “threshold” level [29,30,31].

Vascular endothelium is exposed to the blood-borne FFA, including the albumin-bound FFA and those liberated by the lipoprotein lipase (LPL) from triacylglycerol (TAG)-enriched lipoproteins at the endothelial surface. About a third of LPL-generated FFA is released into circulation in the process called “spillover” [32]. The lipoprotein-associated TAG are markedly higher in obesity [31] and postprandial chylomicrons are the major source of spilled over FFA [33]. Thus, local FFA concentrations at the blood-endothelial interface would be expected to surpass their mean fasting values in plasma and further increase in obesity or postprandial state.

What makes the external FFA detrimental to vascular endothelium in vitro? Low solubility of long-chain FFA in aqueous solutions is the pivotal factor [34]. In blood plasma, FFA are bound to albumin, which serves both the buffering and transporting functions [35]. Albumin concentration in blood plasma is ~0.6–0.7 mM, and in vitro defatted albumin can non-covalently bind up to seven FFA molecules, although with different affinities. Its three high-affinity binding sites cumulatively provide for effective retention of ~2 mM FFA. In blood plasma, albumin can also complex bilirubin, heme, thyroxine, steroid hormones, bile acids, and a wide array of drugs [35]. Perhaps, due to the presence of other ligands, the measured concentration of albumin-bound FFA in human blood rarely exceeds 2 mM. On the other hand, concentration of FFA unbound to albumin stays in the nanomolar range [34], suggesting that once FFAs dissociate from albumin they should promptly complex with other suitable targets including those on/in endothelial cells. Under pathological conditions, the FFA/albumin molar ratio can rise, reaching as high as six to one, unmasking the unbound FFA lipotoxic effects likely mediated by the inflammatory TLR4 receptors and associated signaling [36]. Thus, it is important to use relevant FFA/albumin molar ratios and clearly indicate them in experimental studies; the ratios greater than five should be avoided as they may potentially induce artifacts [34].

## 4. Carnitine Is Indispensable for FFA Utilization and Sequestration

Normally, the bulk of FFA traverses the vascular endothelium for ultimate destination in underlying tissues such as skeletal muscle or adipose tissue. A portion of FFA is retained in endothelial cells and may enter various pathways.

The ability of carnitine to enhance oxidation of fatty acids by vascular endothelium in vitro was demonstrated more than three decades ago [37]. The same authors pointed to the propensity of endothelial cells to loose carnitine in culture [38] (Table 1). Exogenous carnitine increased oleate oxidation and ATP generation in human umbilical artery endothelial cells (HUAEC) cultured in the presence of 0.5 mM oleate [37]. Apparently, rich growth medium used in this study (M199, 20% serum and endothelial cell growth supplement (ECGS) from bovine brain) failed to provide sufficient carnitine to support the carnitine-acyl transferase shuttle to deliver the oleic acid to mitochondria for oxidation (Figure 2). When confirmed by subsequent studies [39,40], the idea emerged that the carnitine loss by cultured endothelial cells may produce an artificial impression that vascular endothelium relies predominantly on glucose and glycolysis for its energy needs.

Carnitine supplementation appears important for all standard growth media (M199, DMEM, MEM, F12, MCDB-131, low-serum EGMs) used to culture endothelial cells in stressful conditions that may critically involve oxidation of fatty acids [39]. As emphasized by Ruderman et al., the “high rates of fatty acid oxidation are demonstrable in HUVEC when they are provided with carnitine” [42]. This is consistent with carnitine acting as a key doorkeeper for entry of fatty acids into mitochondria for oxidation via acyl-carnitine shuttles including the major carnitine-palmitoyl transferase, CPT-1 [43] (Figure 2). This implies that many in vitro studies exploring the effects of FFA on endothelium (Table A2) might have experienced an artificially high rate of apoptosis without carnitine supplementation [28].

The key observation made by Ido et al. [23] indicated that this might be the case, because only in the presence of carnitine, additional AMPK-mediated stimulation of fatty acid delivery to mitochondria rescued the cells from apoptosis. From this and collateral studies [39,40,41], the AMPK-mediated mechanism of endothelial protection was suggested to involve decreased levels of intracellular malonyl-CoA, the inhibitor of CPT-1, and to facilitate carnitine-mediated routing of fatty acids to oxidation in mitochondria [42]. Furthermore, it was suggested that this mechanism is common to both the anti-apoptotic and insulin-sensitizing action of AMPK in hyperlipidemia and hyperglycemia [42,44]. If this mechanism is disabled, an increased glucose supply followed by the accumulation of pyruvate and acetyl-CoA would favor malonyl-CoA production via ACC, formation of DAG, and subsequent activation of PKC, NOX, and generation of ROS (Figure 2) to trigger insulin resistance [45]. Similarly, in the absence of FAO flux, accumulating FFA may trigger inflammatory responses [41] via the long-chain acyl-CoA intermediates [46] and/or oxidative/nitrosative stress leading to apoptosis [47]. Overall, as argued by the studies from the Ruderman lab [23,39,40,41,42] and recently supported by the results of Yao et al. [48], carnitine is key for AMPK to manifest its anti-inflammatory, anti-oxidant, and anti-apoptotic activities in endothelial cells.

Additionally, carnitine may participate in maintaining an equilibrium between the pools of the long-chain acyl-CoA (LCACoA) and free HS-CoA in cells. The intracellular concentrations of LCACoA are normally in the 5–160 μM range but vary considerably in different metabolic conditions [49]. The excess of LCACoA is potentially injurious as it might produce a spectrum of aberrant metabolic and signaling switches [46,49] leading to insulin resistance, inflammatory response, dysregulated autophagy, oxidative stress, and apoptosis. Many of these features are observed when endothelial cells are subject to experimental hyperglycemia (Table A1) or hyperlipidemia (Table A2), with carnitine being protective in these settings (reviewed in [50]).

Carnitine is synthesized mainly in the liver and distributed through the blood to other tissues including vascular endothelium. With carnitine concentration in the blood plasma at ~50 μM, it is expected to be about the same in endothelium that is constantly exposed to blood. This is in the range of free and acyl-bound CoA. Thus, due to intrinsic ability to reversibly exchange the acyl groups with many acyl-CoA species ranging from simple acetyl-CoA to LCACoA, carnitine may be essential to maintain the availability of the HS-CoA pool for metabolic needs and to sequester excess LCACoA (Figure 2) in metabolic disorders such as obesity and T2D.

## 5. AMPK Controls the Malonyl-CoA-Dependent Checkpoint of FFA Utilization

The original discovery that, in addition to being a precursor for the de novo synthesis of fatty acids, malonyl-CoA serves as an acyl-carnitine transferase inhibitor [51] (see [46,52] for explicit reviews), has eventually developed into a key molecular mechanism, mediated by AMPK, that controls both fuel switching by cells and emergence of insulin resistance [42,44,53,54]. Through ACC phosphorylation, AMPK inhibits malonyl-CoA synthesis from acetyl-CoA thereby limiting accumulation of DAG and ceramides that induce ROS generation and apoptosis. Simultaneously, AMPK relieves CPT-1 inhibition by malonyl-CoA. This allows the fatty acid influx into mitochondria (Figure 2), peroxisomes, and endoplasmic reticulum for catabolism [52]. In addition, the skeletal muscle malonyl-CoA-decarboxylase (MCD), which limits the malonyl-CoA amount available for LCACoA formation, is also likely to be under AMPK control [55]. It is unknown whether AMPK similarly regulates MCD and LCACoA levels in vascular endothelium. Despite several other steps of the prototype AMPK/ACC/MCD regulatory mechanism remain to be demonstrated in endothelial cells, it is consistent with the notion that AMPK activity is reduced in diabetic conditions [21,22] and that upregulated FAO may rescue endothelial cells from apoptosis [37,39,48].

Based on the outlined metabolic relationships (Figure 2), it is anticipated that under conditions of excessive glucose or FFA and lack of carnitine and insufficient activity of AMPK in cultured endothelial cells, multiple mechanisms may contribute to endothelial apoptosis. They likely include increased auto-, mito-, or lipophagy [56,57,58,59], the polyol, AGE, and hexosamine pathways branched from glycolysis [20,60], inflammatory signaling and cytokines [61,62], or oxidative/nitrosative stress mediated by NOX or eNOS [47,63,64,65,66,67,68].

In the absence of carnitine and/or with low AMPK activity, the increased malonyl-CoA would be likely rerouted toward de novo synthesis of palmitate, DAG/TAG, and lipid droplet formation. Avoiding the risk of insulin resistance can be achieved via lipid droplet utilization by auto(lipo)phagy. This appears to occur in cells treated with exogenous fatty acids [56,58,59] or high glucose [57]. Inhibiting the early steps in autophagosome formation by 3-methyladenine (3-MA) blocks the Nox4-mediated ROS accumulation, the activating phosphorylation of PKCa [56], and, most importantly, apoptosis [58]. Silencing of vps34, a putative target of 3-MA, produces the same effect [58]. While the mechanism by which auto(lipo)phagy activates the Nox4/ROS response is still elusive, ROS involvement in apoptosis is well conserved in vascular endothelial cells (Table A1 and Table A2).

Whether and to what extent the above-mentioned mechanisms are realized in the presence of carnitine and sufficient AMPK activity and lead to endothelial cell dysfunction and damage needs to be re-evaluated.

## 6. Mitochondrial Uncoupling as ROS Reduction Option: Potential Mechanisms

Mitochondria are considered the major source of superoxide anion and other ROS, at least in hyperglycemia [65,69,70]. ROS accumulate as by-products of intense aerobic metabolism when the mitochondrial electron transport chain is overloaded and the inner membrane potential increases [57,71]. While there are many multifaceted relationships between mitochondrial ROS and programmed cell death [72], lowering the mitochondrial membrane potential is thought to decrease the production of superoxide anion radicals and to prevent apoptosis. The mitochondrial uncoupling protein UCP-2 belonging to the UCP family [73] was identified in various tissues including vascular endothelium [74,75]. According to suggested models, a member of the UCP family, UCP-1, allows FFA anions to flip from mitochondrial matrix to the outer side of inner mitochondrial membrane where FFA anions are neutralized by accumulated protons and diffuse back into the matrix, where they release protons. This mechanism dissipates the proton gradient (Δμ_H+_) across the inner mitochondrial membrane and reduces ATP synthase activity [76]. In mammals, UCP-2 also reduces the mitochondrial membrane potential, attenuates mitochondrial ROS production, and protects against oxidative damage [73]. Protective role of mitochondrial uncoupling has been demonstrated by UCP overexpression in endothelial cells. Forced expression of UCP-1, normally found in brown and beige adipocytes, rescues the endothelial cells from high glucose-induced apoptosis [70]. Overexpression of UCP-2 in HAEC inhibits linoleic acid-induced ROS production, NF-kB activation, and apoptosis [77]. High glucose and fatty acids upregulate endogenous UCP-2 expression [73,78], likely demonstrating an adaptive response of endothelial cells to higher mitochondrial respiration. Noteworthy, UCP-2 expression/activity as well as mitochondrial biogenesis stay under AMPK control through the transcriptional peroxisome proliferator coactivator PGC-1a, thus reducing ROS generation by mitochondria in the vascular endothelium [79,80]. In turn, AMPK activity appears to be upregulated by ROS, thus forming a positive feedback circuit to dump mitochondrial ROS production and prevent oxidative stress [68,79]. By this virtue, AMPK seems to function as a dual sensor of both the energy and redox status within a cell [81]. All these findings substantiate an idea that the reduced AMPK activity in prediabetic or diabetic states may aggravate mitochondrial dysfunction and cause insulin resistance [53].

There is also another potential mechanism that may rescue the mitochondrial electron transport chain from overloading. Inability of a cell to spend ATP increases the mitochondrial ROS; restoring the ATP generation lowers the proton electrochemical potential gradient (Δμ_H+_), increases oxidation of electron carrier pools, and reduces local oxygen to cease superoxide anion radical production in mitochondria [69,82]. Such an ATP-spending mechanism was originally found in neuronal cells and appears to involve either mitochondria-bound hexokinase [83] or creatine kinase [84]. It allows ATP to recycle, which may provide the “mild” uncoupling of mitochondria to prevent ROS formation [82,85]. An increased expression of hexokinase-I has been observed in EA.hy926 endothelial cells treated with high glucose [78]. Whereas it is the hexokinase-II isoform that is generally thought to be mitochondria-targeted, in fact, both isoforms are capable of mitochondria binding and causing “mild” uncoupling [85]. Notably, this mechanism has been implicated as an inherited component of the whole-body anti-aging program [85], but whether it upregulates in endothelial cells to counter the ROS-induced apoptosis is yet unknown.

Similarly, the creatine system (reviewed by [86]) shares much in common with carnitine and the hexokinase mechanism. Endothelial cells contain both creatine transporter (SLC6A8) and substantial amounts of mitochondria-bound creatine kinase [87]. The creatine concentration in the blood plasma is normally ~50 μM but in the typical endothelial EBM-2 culture medium (Bio-Whittaker Inc., USA) containing 5% fetal calf serum, it is ~20-fold less [88]. This suggests that, as with carnitine, the cellular creatine stores may exhaust upon culturing.

A number of studies have elaborated the antioxidant potential of creatine [86], however only a few have addressed whether it controls the mitochondrial membrane potential and ROS production. When human pulmonary endothelial cells were cultured with up to 5 mM extra creatine, the levels of intracellular creatine and phosphocreatine proportionally increased, but the ATP levels remained constant suggesting that mitochondrial ATP generation exceeded initially limited values [88]. Remarkably, this extra creatine inhibited ICAM-1 and E-selectin expression, neutrophil adhesion, and tightened the endothelial barrier by reducing its transient permeability in response to serotonin or hydrogen peroxide, all indicative of the anti-inflammatory effect of creatine. This was substantiated by Meyer et al. in neuronal cells, who directly showed that the ATP⇄ADP recycling by mitochondria-bound creatine kinase decreases ROS generation particularly under high glucose conditions [84]. Studies in cultured HUVEC and C2C12 myoblasts confirm that creatine supplementation protects cells from an oxidant-induced injury and increases cell viability [89,90]. Mechanistically, these effects of creatine could relate to its capability to activate AMPK, PGC-1a, and mitochondrial biogenesis [89].

## 7. Conclusions

Summing up, the in vitro endothelial models require a thoughtful approach with regard to essential natural components of the culture medium to be available for endothelial cells in order to cope with external metabolic interrogations, such as those occurring in diabetic states. Both high glucose and increased FFA load alter the cellular metabolism to increase intracellular malonyl-CoA and LCACoA and thrust them into the cytosolic lipogenesis flux and/or into mitochondria for the β-Ox/FAO and OxPhos pathways (Figure 2). The choice and the balance are determined by the activities of the rate-limiting enzymes and/or their regulators. Carnitine seems essential for LCACoA buffering and transferring to mitochondria; the lack of carnitine function increases the risk of cell apoptosis. Easy loss and uptake of carnitine appears to be habitual for endothelial cells; therefore, it is essential to maintain endothelial cultures with carnitine supplements, especially in metabolic studies. Unless it is done, LCACoA would be likely rerouted to the lipogenesis pathway, increasing the risk of lipid droplet formation, PKC-mediated insulin resistance and NOX activation, ROS generation, and apoptosis. This flux may also involve the auto(lipo)phagy, which also results in ROS generation by an unknown mechanism(s).

Under high glucose conditions, an increased glycolytic flux triggers three major pathways involved in endothelial cell damage: through polyol/sorbitol or G3P-initiated AGE formation, or F1,6PP-initiated hexosamine pathway (Figure 2). It also increases the levels of dihydroxyacetone phosphate (DHAP) and malonyl-CoA, resulting in both increased de novo synthesis of LCACoA and the lipogenesis flux with all detrimental consequences discussed above.

Involvement of ROS in endothelial dysfunction in diabetic conditions is hardly disputed and seems to be a unifying pathogenic mechanism [14,20]. Yet, differences in the rates of apoptosis in endothelial cell cultures and in the vessels in situ raise the possibility that ROS mechanisms may differ in their extent or contribution. Even further, the ROS-mediated apoptosis may not be the bona fide culprit of diabetic endothelial dysfunction but rather an in vitro side-effect that reflects a primary defect, which may not culminate to such an extreme in vivo until the terminal stages of the disease.

Is there a safeguard regulatory circuit present in cultured/diabetic endothelial cells to protect from detrimental ROS generation? AMPK appears to be an important molecular component of such a circuit. AMPK is an intrinsic double-faced regulator of the cytoplasmic malonyl-CoA and LCACoA levels and mitochondrial ROS (Figure 2). AMPK exerts double control of malonyl-CoA production through ACC inhibition and MCD activation in cytoplasm; AMPK blocks ROS generation through increased mitochondrial biogenesis and respiration. The latter mechanisms are not fully understood but likely include activation of PGC-1α and upregulation of UCP-2. An intriguing question remains as to whether these AMPK effects on ROS involve ATP recycling via mitochondria-bound hexokinase and/or creatine kinase in endothelium. Accumulated data outlined in this review bestow on AMPK a likely potential to be a key regulator of the ROS status in endothelial cells. AMPK activity is lower in insulin-resistant and diabetic states [22,53], making it a promising therapeutic target.

In conclusion, the endothelial cell-based models appear indispensable to explore cellular mechanisms of vascular dysfunction in diabetes-related conditions such as hyperglycemia and/or dyslipidemia. For metabolic studies, the cells need to be constantly supplied with essential factors. Some of them may not yet be fully elaborated, likely because the harsh metabolic conditions had not been envisioned when common cell media were being developed. As evidence accumulates, these essentials may be needed to revive the intrinsic safeguard mechanisms, such as those involving AMPK, to sustain mitochondrial function, combat oxidative stress and prevent artificially high apoptosis. This allows maintaining the cultures for up to several weeks to monitor the long-term progression of cell metabolic dysfunctions. Still, endothelial cell-based models retain limitations with regard to systemic responses imposed by hemodynamic forces or communication with other cells, including blood cells or vascular wall neighbors. These aspects can be also addressed in advanced coculture systems. Clearly, more studies are needed, under the controlled cell culture conditions, to reap the harvest.

## Figures and Tables

**Figure 1 biomedicines-10-03181-f001:**
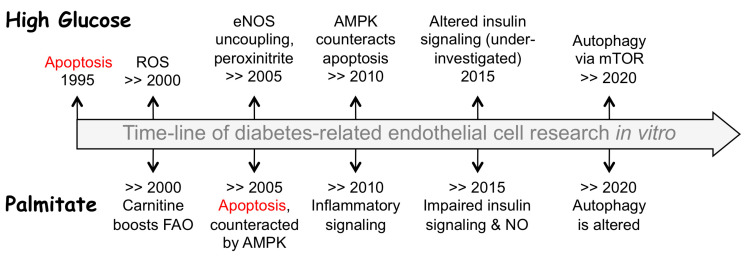
Schematic view of research progress using endothelial cell-based models of hyperglycemia (high glucose, top) or hyperlipidemia (palmitate, underneath). The legends next to the years depict major advances in identifying the key detrimental events and molecular players further detailed in the text. Abbreviations: AMPK, AMP-activated protein kinase; eNOS, endothelial NO-synthase; FAO, fatty acid oxidation; mTOR, mammalian target of rapamycin; NO, nitric oxide; ROS, reactive oxygen species.

**Figure 2 biomedicines-10-03181-f002:**
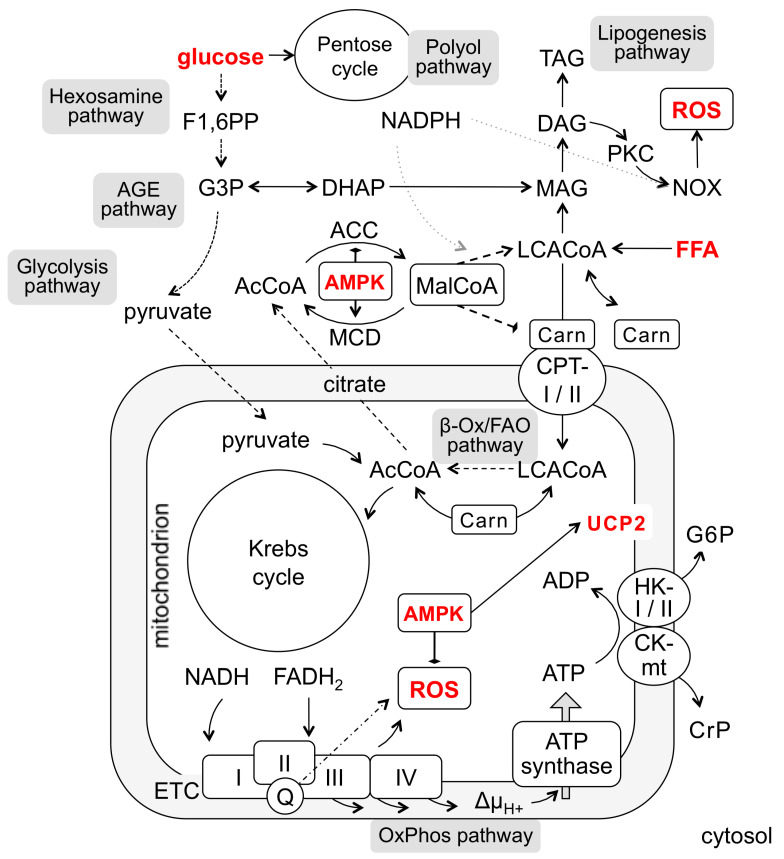
Endothelial metabolic wires altered by hyperlipidemia and hyperglycemia. The major pathways are indicated as grey boxes. Molecular actors in main focus are marked in red. The scheme embraces cytosolic carbohydrate metabolism (top left) and fatty acid turnover (top right); utilization of these substrates in mitochondria for energy production is boxed underneath. Abbreviations: ACC, acetyl-CoA carboxylase; AcCoA, acetyl-coenzyme A; ADP, adenosine diphosphate; AMPK, AMP-activated protein kinase; ATP, adenosine triphosphate; Carn, carnitine; CK-mt, mitochondrial isoform of creatine kinase; CPT-1, carnitine-palmitoyl transferase; CrP, creatine phosphate; DAG, diacylglycerol; DHAP, dihydroxyacetone phosphate; ETC, electron transport chain of mitochondria; F1,6PP, fructose 1,6 biphosphate; FADH2, flavin adenine dinucleotide reduced; FAO, fatty acid oxidation; FFA, free fatty acid; G3P, glyceraldehyde 3-phosphate; G6P, glucose-6-phosphate; HK I/II, hexokinase I/II isoform; LCACoA, long-chain acyl-coenzyme A; MAG, monoacylglycerol; MalCoA, malonyl-coenzyme A; MCD, malonyl-CoA decarboxylase; NADH, nicotinamide adenine nucleotide reduced; NADPH, nicotinamide adenine nucleotide phosphate reduced; NOX, NADPH-oxidases; OxPhos, oxidative phosphorylation; PKC, protein kinase C; Q, coenzyme Q, ubiquinone; ROS, reactive oxygen species; TAG, triacylglycerol; UCP-2, uncoupling protein 2; β-Ox, β-oxidation of fatty acids; Δμ_H+_, membrane electrochemical potential; and I, II, III, IV, complexes I–IV of ETC.

**Table 1 biomedicines-10-03181-t001:** Effects of carnitine on endothelial cells in vitro.

Cells	Culture Conditions	FFA Duration and Concentrations, Other Essentials	Assessed Parameters	Major Findings/Mechanism	Reference
HUAEC (arterial)	M199, 10% FCS, 10% HS, ECGS, passage 2–4	0.5 mM oleate, 50 μM carnitine	FAO, ATP generation	Carnitine stimulates FAO and ATP production by ~2.5-fold	[37]
HUVEC	EBM-2 (Clonetics), 50 μM carnitine (in metabolic studies), passage 3–5	0.11 mM palmitate in metabolic studies; 0.2–2 mM AICAR, 0.5–2 h	FAO vs. Glucose metabolism	In the presence of carnitine, AMPK activation by AICAR reverts energy production from glycolysis to FAO	[39]
HUVEC	EBM-2 (Clonetics), 50 μM carnitine, passage 3–5	0, 5, or 30 mM glucose, 2 h; 0.1 mM palmitate, 24 h in metabolic studies	FAO vs. Glucose metabolism	AICAR reduces malonyl-CoA and glycolysis, but increases FAO ~3-fold only in the presence of carnitine	[40]
HUVEC	EGM-2 (Clonetics) (for cell growth); M199, 10% FBS (for assays); 50 μM carnitine (in metabolic studies), passage 5–6	5 vs. 30 mM glucose, 24–72 h, 1 mM AICAR, 24 h; 0.1 mM palmitate, 24 h in metabolic studies	Apoptosis (TUNEL), caspase-3, P-Akt (S473, WB), FAO vs. glucose oxidation, ATP, DAG, ceramide, malonyl-CoA	AICAR or AMPK forced expression inhibits apoptosis, decreases malonyl-CoA and DAG, increases FAO and P-Akt, reduces lactate, pyruvate, and glucose oxidation	[23]
HUVEC	EBM-2 (Cambrex), M199, 5% FBS, 50 μM carnitine with FFA, passage 4–6	5 vs. 25 mM glucose, 24 h, 0.4 mM palmitate, 16–24 h	NF-kB, VCAM-1 expression	Palmitate (but not HG) increases, but AICAR decreases both NF-kB activation and VCAM-1 expression	[41]

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
