# Peer review of "In Vitro Modeling of Diabetes Impact on Vascular Endothelium: Are Essentials Engaged to Tune Metabolism?"

_biomedicines, 2022, doi:10.3390/biomedicines10123181_

Round 1
Reviewer 1 Report
The review summarizes the in vitro study to mimic hyperglycemia and hyperlipidemia, which are hallmarks of type 2 diabetes and discusses the role of carnitine, creatine, and AMP-activated protein kinase in endothelial metabolism. The entire review is well-organized and addresses significant points. However, I have a couple of several suggestions.
1. subtitle for each section is better to not only reflect what aspect is talked about but also it is necessary to address the primary function or the central point for the function. The subtitle should contain more information.
2. Tables 1 and 2 in sections 2 and 3 are better to re-cluster. These two tables contain comprehensive information about high glucose/ FFA outcomes on endothelial cells. It is better to summarize the cell & study design, culture conditions, glucose/FFA duration & concentrations, other essentials, and assessed parameters into categories. It may make the table easy to read.
Author Response
We appreciate the reviewer comments and suggestions to improve the manuscript. The text changes we made are highlighted in yellow. We also restructured and somewhat text-edited the appendix tables, which changes are not highlighted to avoid variegation.
Reviewer 1
- subtitle for each section is better to not only reflect what aspect is talked about but also it is necessary to address the primary function or the central point for the function. The subtitle should contain more information.
Response
We revised the subtitles throughout the manuscript and made them more informative.
- Tables 1 and 2 in sections 2 and 3 are better to re-cluster. These two tables contain comprehensive information about high glucose/ FFA outcomes on endothelial cells. It is better to summarize the cell & study design, culture conditions, glucose/FFA duration & concentrations, other essentials, and assessed parameters into categories. It may make the table easy to read.
Response
We purposefully placed Tables A1 and A2 in appendix section in contrast to Table 1 (carnitine) left in the main text. We realized that bulky size and detailed content of these tables would impede reading of the review. On the other hand, these tables allow readers interested in details of a particular study to quickly get its digest and follow to the original paper. For those readers who prefer to get a snapshot of a problem all tables are distilled in Figure 1 and in the main text.
Prompted by the reviewer suggestion we tried to re-cluster Tables A1 and A2 but found it difficult to achieve any easy reading format without considerable loss of descriptive details that justify the very existence of these tables as the knowledge database. We finally decided on a compromise. We re-arranged tables into two major categories by the duration of glucose / FFA treatment. We set the dividing margin to 2 days for palmitate and 3 days for glucose, based on the most of used protocols and our own observations. The result did make some sense, so we further sub-clustered studies according to whether apoptosis or just associated events were assessed (the empty or shadowed rows as before). In addition, we slightly text-edited the Table contents and included a new 2021 paper (Hansen et al., 2021) missed from the original data analysis. It went into both tables as both the HG and palmitate treatments were performed in that study. We feel that the Tables became upgraded, but could not be further compacted without loss of valuable information. We modified the Table titles accordingly.

Reviewer 2 Report
The study of Vorotnikov and colleagues offers a comprehensive overview about this topic.
In my opinion three important aspects were not sufficiently addressed:
The authors should discuss how the duration of the interventions might influences the reported results (the longest reported intervention was 7 days – diabetes lasts for years). Limitations in this regard and how to overcome them.
The authors nicely collected relevant interventions but might also include recommendations for best laboratory practice (please specify your conclusions).
Since acute vascular complications (e.g. atherothrombotic events) are clinically important, the authors might include a section in which they discuss relevant literature.
Author Response
We appreciate the reviewer comments and suggestions to improve the manuscript. The text changes we made are highlighted in yellow. We also restructured and somewhat text-edited the appendix tables, which changes are not highlighted to avoid variegation.
Reviewer 2
- The authors should discuss how the duration of the interventions might influences the reported results (the longest reported intervention was 7 days – diabetes lasts for years). Limitations in this regard and how to overcome them.
Response
In fact, this is the major point throughout the review, which we discuss all over. We are convinced that it is the absence of some essential natural components in the commonly used endothelial cell growth media that makes the cells vulnerable to stringent 'metabolic' conditions. Artificially high rate of endothelial apoptosis reported in the multiple studies especially in the presence of FFA simply prevented long-term studies of hyperlipidemia on vascular endothelial cells in vitro and shaped erroneous opinion in literature regarding acute lipotoxic effects of FFA and palmitate, in particular, on endothelium. Our recent study (Samsonov et al., 2021) appeared among a few to demonstrate that under biochemically proper culture conditions palmitate interferes with insulin signaling in endothelial cells but does not bring cells to death for several weeks of experiments. We believe that it is more adequate in vitro representation of in vivo situation in diabetic vasculature but admit that it is still a limited cell-based in vitro model that, however, has potential for further improvement. We offer to interested researches our experimentally supported recommendations for setting up a long-term endothelial cell culture in diabetic conditions and suggest that many previous studies should be re-evaluated based on this methodology. We added the brief summing message as the last para in the Conclusions section where we also noted that those missed essentials are likely to work in cooperation with AMPK. We also hint at the hemodynamic forces as the likely natural stimulus keeping AMPK active in vascular endothelium even in the presence of diabetes.
- The authors nicely collected relevant interventions but might also include recommendations for best laboratory practice (please specify your conclusions).
Response
Perhaps, the previous response relates to this comment as well. The introductory “recipe” for setting up a long-term human vascular endothelial cell culture in hyperlipidemic conditions may be found in (Samsonov et al., 2021). This paper is open to public. The importance of the proper preparation of FFA:albumin complex is discussed in the review. Currently, we are not ready to present the complete list of SOPs to support diabetic / metabolic studies in long-term endothelial cell-based culture systems. It is under development by us and hopefully by others and requires further research.
- Since acute vascular complications (e.g. atherothrombotic events) are clinically important, the authors might include a section in which they discuss relevant literature.
Response
We located the suitable place for such a section in the Introduction. We added the phrase describing association of clinical diabetes and atherosclerosis and very briefly outlined pathologic processes that lead to vascular damage, atherothrombosis and increased cardiovascular mortality typical for this comorbidity. For the detailed description of these issues, we provided references to published review articles that highlight various aspects of clinical / experimental studies of vascular dysfunction in the context of diabetes and atherosclerosis. Alterations made are marked in yellow.
